# Pychastic: Precise Brownian Dynamics using Taylor-Itō integrators in Python

Radost Waszkiewicz[1]*, Maciej Bartczak[1], Kamil Kolasa[1], and Maciej Lisicki[1]

[1] Institute of Theoretical Physics, Faculty of Physics, University of Warsaw
L. Pasteura 5, 02-093 Warsaw, Poland
* radost.waszkiewicz@fuw.edu.pl

December 20, 2022

## Abstract

In the last decade, Python-powered physics simulations ecosystem has been growing steadily, allowing greater interoperability, and becoming an important tool in numerical exploration of physical phenomena, particularly in soft matter systems. Driven by the need for fast and precise numerical integration in colloidal dynamics, here we formulate the problem of Brownian Dynamics (BD) in a mathematically consistent formalism of the Itō calculus, and develop a Python package to assist numerical computations. We show that, thanks to the automatic differentiation packages, the classical truncated Taylor-Itō integrators can be implemented without the burden of computing the derivatives of the coefficient functions beforehand. Furthermore, we show how to circumvent the difficulties of BD simulations such as calculations of the divergence of the mobility tensor in the diffusion equation and discontinuous trajectories encountered when working with dynamics on $S^2$ and $SO(3)$. The resulting Python package, Pychastic, is capable of performing BD simulations including hydrodynamic interactions at speeds comparable to dedicated implementations in lower-level programming languages, but with a much simpler end-user interface.

# 1  Introduction

The dynamics of soft and colloidal matter systems is of importance for numerous technological and industrial processes, such as food products, pastes, creams, and gels [2]. Another important example are biological systems which involve aqueous suspensions of colloids, macromolecules, polymers, and cells. The diversity of the constituent elements, together with the tunability of their direct interactions, and the presence of hydrodynamic interactions (HI) mediated by the suspending fluid [3], gives rise to a multitude of complex dynamic phenomena, which can only be explored using appropriate numerical techniques [4, 5].

The choice of a suitable method derives from the characteristic time scales of the investigated dynamics [6]. In the context of macromolecules, when short-time effects are of interest, Molecular Dynamics (MD) simulations are a popular choice [7]. Short times may be comparable to the solvent relaxation time scale, which for a fluid of kinematic viscosity $v$ and speed of sound $c$ scales as $v/c^2$. For water, this time is of the order of $10^{-13}$ s. The idea behind MD simulations is to solve Newtonian equations of motion for atoms or molecules, which are a set of second-order ODEs with effective interaction potentials. In this case, HI can be resolved using either explicit solvent methods that resolve the molecular structure of the solvent or more approximate implicit solvent models. However, the typical time scales of colloidal motion are much longer; the velocity relaxation for colloidal particles of size $a$ and density comparable to that of the fluid is approximately $a^2/v$, amounting to ca. $10^{-8}$ s for a 100 nm colloid. On coarser time scales, the colloidal velocity relaxes multiple times, and the Rayleigh description in terms of velocity becomes irrelevant [6, 8]. Instead, the description in terms of the position of a Brownian particle becomes possible on diffusive time scales, during which a particle diffuses over a distance of its own radius. This time scale is $a^2/D \sim \eta a^3/k_B T$, where $\eta$ is the dynamic viscosity of the fluid, $k_B$ is the Boltzmann constant, and $T$ is the temperature. For the aforementioned particle, this time scale is ca. $10^{-3}$ s, and corresponds to the minimal time resolution of typical light scattering and microscopy experiments. Thus, the description of the dynamics on such coarse scales is made solely using the position of the particle. The clear separation of discussed time scales, combined with the numerical stiffness of Newton's equations, renders MD simulations not applicable to explore the diffusive dynamics.

Instead, BD simulations present a convenient alternative, building on models suitable for diffusive time scales, and involving accordingly long time steps for numerical computations. Since the velocity of particles varies very rapidly on such time scales, the position becomes a random variable whose properties are dictated by the fluctuation-dissipation theorem [8].

The development of a suitable theoretical description to explain the Brownian motion and its subsequent experimental verification at the beginning of the 20th century were a groundbreaking step that confirmed the atomic structure of matter [9]. Early works by pioneers such as Ein-

stein [10], Smoluchowski [11], Fokker [12], and Planck [13], led to the description of diffusive processes through the underlying probability density functions (PDFs) for the positions of the particles and their temporal evolution, rather than representations of individual trajectories. The realisation that the path of a Brownian particle resembles a nowhere differentiable function paved the route for a different approach, now called stochastic calculus, with the classical works of Kolmogorov [14], Wiener, and finally Itō [15]. With these tools, the problem of Brownian motion can be recast in the form of stochastic differential equations (SDEs), which offer a practical route for numerical simulations.

In this contribution, we present our approach to implementation of such Brownian dynamics. Our goal was to show how modern programming techniques available in Python: reflective and functional programming paradigms allow one to develop concise implementations of physically relevant problems. These constitute relatively recent developments in computer modelling, and we follow this trend to facilitate faster modelling of diffusive phenomena.

First, in Sec. 2 we review three popular ways of describing the Brownian motion using the Langevin equation, the Fokker-Planck equation, and Itō equation. We then provide an overview of available numerical integration packages for SDEs in Sec. 3.1. Next, we present our new package, Pychastic, in Sec. 3, along with three examples of problems in which the packages greatly facilitate numerical computations. We conclude the work in Sec. 5.

## 2 Three vantage points: Langevin, Fokker-Planck, and Itō

Perhaps the most popular approach to modelling the trajectories of Brownian particles dates back to Langevin's work of 1908 [16,17]. The Langevin equation can be rationalised as an extension of Newtonian mechanics to include the effects of fluctuations by adding a stochastic force, $F_n$, acting on a particle. We write it below for a particle with a constant friction coefficient $\zeta$. The average value of the force is zero, while its covariance at temperature $T$ is given by the fluctuation-dissipation theorem [8], $\langle F_n(t)F_n(t')\rangle = 2k_B T\zeta\delta(t-t')$, with $k_B$ being the Boltzmann constant. The equation of motion of the particle in the presence of a deterministic force $F$ reads then

$$m\ddot{x} = -\zeta\dot{x} + F(x,t) + F_n(t), \tag{1}$$

For the case of a spherical particle of radius $a$ in Stokes flow the friction coefficient is $6\pi\eta a$, $\eta$ being the dynamic viscosity. We note that even though $x(t)$ is not differentiable once, and certainly not differentiable twice at any point even in the usual distributional sense (because the Lebesgue integral requires finite variation and realisations of the Wiener process have infinite variation), Eq. (1) can be given a proper interpretation by transforming it into an integral form. However, the above equation proved practical for numerical calculations, e.g. using an integration scheme analogous to the Euler-Maruyama method.

An alternative description via the Fokker-Planck equation circumvents the difficulties of interpreting the dynamics on a single trajectory in the Langevin equation by focussing on a probabilistic description. On the Brownian time scale, the position of a particle is now treated as a random variable, and the probability density function (PDF) $P(x,t)$ of finding the particle at a location $x$ at a time $t$ evolves according to the partial differential equation

$$\frac{\partial P(x,t)}{\partial t} = \zeta^{-1}\frac{\partial}{\partial x}\left(F(x,t)P(x,t)\right) + D\frac{\partial^2 P(x,t)}{\partial x^2}, \tag{2}$$

with the diffusion coefficient defined by $D = k_B T/\zeta$. The problem is now well-posed mathematically, and the tools of mathematical analysis of PDEs can now be employed to study the evolution of the underlying probability distribution, e.g. by examining its moments. However, questions involving individual trajectories become less straightforward: problems involving first passage time have to be dealt with with a careful treatment of the boundary conditions (e.g. substances vanishing in chemical reactions). Moreover, from the point of view of numerical simulations, this approach becomes impractical for large systems. Using finite-difference methods, for $N$ particles in a 3D simulation box of size $L$ and mesh size $\Delta x$, we typically require $(L/\Delta x)^{3N}$ points to track the PDF, which may become prohibitively large.

The mathematical difficulties of the Langevin equation are absent in the proper treatment of SDEs in the Itō formalism. If now $dX$ denotes the position increment of a particle in a time interval $dt$, we can write it as

$$dX = F dt + \sqrt{2D}dW, \tag{3}$$

where $F dt$ denotes the systematic drift of the particle and $\sqrt{2D}dW$ denotes the stochastic (diffusive) component of motion (provided that $D$ is constant and the metric tensor is constant as well). This equation is meant in the distributional sense with respect to the Itō integral, that is

$$X(T) = \int_0^T dX = \int_0^T F(X,t)dt + \int_0^T \sqrt{2D}dW. \tag{4}$$

Here, all integrals are taken in the Itō sense. Such a trajectory-focused formulation effectively deals with the ill-defined derivatives in the Langevin equation. Using Itō's lemma imposes formal rules of transformation of the coefficients, freeing us from a canonical coordinate description. Finally, the estimation of observable quantities such as expected values, correlation functions or equilibrium distributions of low-dimensional projections of evolving variables can be recovered using a Monte Carlo approach, which for $M$ simulations converges as $\sqrt{M}$, regardless of dimensionality. The constant of proportionality of this convergence is controlled only by the variance of the variable of interest, which is often independent of the dimensionality of the equation. For example, in the problem of diffusion of a single particle in a semidilute suspension, the variance of the mean squared displacement of the tracer is weakly dependent on the number of particles in the simulation volume.

## 3 Pychastic: description of the package

### 3.1 Available numerical integration packages

Only a few SDE integration packages have been made available in recent years. Two notable examples are `DifferentialEquations.jl` for Julia and `ItoProcess` being a part of Mathematica. Now, we present `Pychastic` for Python, which takes advantage of the popularity of this language. Our package source code is available on Github, up-to-date documentation on ReadTheDocs and ready to install via `pip` via Python Package Index. As a preliminary comparison, we note that `DifferentialEquations.jl` has the largest variety of integrators (for example, many options for stiff equations), while Mathematica's `ItoProcess` and `Pychastic` contain essentially the same algorithms. However, the biggest drawback of `ItoProcess` is the lack of step post-processing. When working with SDEs defined on manifolds whose universal cover is not $\mathbb{R}^n$, such as a sphere $S^2$ or the space of rigid rotations $SO(3)$, any parameterisation of $\mathbb{R}^n$ will contain singularities, thus an

| Package | DifferentialEquations.jl | ItoProcess | Pychastic |
|---|---|---|---|
| Language | Julia | Mathematica | Python |
| License | MIT | proprietary | MIT |
| Codebase | open | closed | open |
| scalar SDEs | yes | yes | yes |
| vector SDEs | yes | yes | yes |
| strong convergence | up to order 1.5 | up to order 1.5 | up to order 1.5 |
| weak convergence | up to order 2.0 | up to order 2.0 | up to order 2.0 |
| supports events | yes | no | yes |

Table 1: Available SDE integration packages.

integrator which cannot handle discontinuous paths cannot reproduce, e.g. 3- dimensional rotational dynamics [18]. Table 1 contains a synthetic comparison of the three mentioned packages.

## 3.2 Implementation details

`Pychastic` contains implementations of three numerical integration schemes based on the Taylor-Itō expansion. They are the schemes of strong order 1/2, 1, and 3/2 which in the package are referred to as Euler, Milstein, and Wagner-Platen schemes, respectively.

The basis for these integration schemes is the Taylor-Itō expansion [1] which generalises the deterministic Taylor expansion to SDEs. We write it here for a one-dimensional problem driven by a Wiener process $W$

$$dX = a(X)dt + b(X)dW. \tag{5}$$

Following the notation of Kloeden & Platen [1], the Taylor-Itō expansion up to strong order 3/2 for a scalar function $X$ has the form

$$
\begin{aligned}
X(T) =& X(0) + a\Delta + b\Delta W \\
&+ \frac{1}{2}bb'\big((\Delta W)^2 - \Delta\big) \\
&+ a'b\Delta Z + \frac{1}{2}\Big(aa' + \frac{1}{2}b^2 a''\Big)\Delta^2 \\
&+ (ab' + \frac{1}{2}b^2 b'')(\Delta W\Delta - \Delta Z) + \frac{1}{2}b\big(bb'' + (b')^2\big)\Big(\frac{1}{3}(\Delta W)^2 - \Delta\Big)\Delta W,
\end{aligned}
\tag{6}
$$

where

$$\Delta = \int_0^T dt = T, \tag{7}$$

$$\Delta W = \int_0^T dW = W(T) - W(0), \tag{8}$$

$$\Delta Z = \int_0^T \int_0^s dW(s)\,ds. \tag{9}$$

The stochastic Euler scheme is based on the first line of this expansion, with three terms only. The Milstein scheme includes the next term, namely the second line of Eq. (6). Finally, the Wagner-Platen scheme includes all the terms mentioned above. We note from this expansion that even the

simplest schemes require correct computation of coefficients multiplying principal Wiener integrals ($\Delta$, $\Delta W$, $\Delta Z$, and others in the multidimensional case) and sampling from correct distributions corresponding to these integrals. Importantly, an analogous expansion for vector quantities is considerably more complex, and other Wiener integrals arise, as detailed in [1]. For brevity, we do not write this expansion here explicitly, but we have implemented the vector Taylor-Itō expansion in Pychastic to enable simulations of both scalar and vector processes.

Although expressions for these coefficient functions can be written explicitly in principle, in Pychastic we take advantage of functional programming tools, which results in a greatly simplified implementation. First, by introducing the $\mathcal{L}^0$ and $\mathcal{L}^j$ operators (again using the Kloeden & Platen notation), we can express all coefficient functions by repeated application of the $\mathcal{L}$ operators to the $a$ and $b$ functions. Using the jax.grad functionality, this is implemented directly as

```python
def tensordot1(a, b):
    return jax.numpy.tensordot(a, b, axes=1)

def tensordot2(a, b):
    return jax.numpy.tensordot(a, b, axes=2)

# Taylor-Ito expansion operators
def L_t_operator(f,problem):
    @wraps(f)
    def wrapped(x):
        b_val = problem.b(x)
        val = tensordot1(jax.jacobian(f)(x), problem.a(x)) + 0.5 * tensordot2(
            jax.hessian(f)(x), tensordot1(b_val, b_val.T)
        )
        return val[:,jnp.newaxis,...] #indexing convention [spatial, time, ... =
    noiseterms/time]

    return wrapped

def L_w_operator(f,problem):
    @wraps(f)
    def wrapped(x):
        val =  tensordot1(jax.jacobian(f)(x), problem.b(x))[:,jnp.newaxis,...]
        return jnp.swapaxes(val,1,-1)[:,...,0] # indexing convention [spatial,
    noiseterms, ... = noiseterms/time]

    return wrapped
```

Second, different integration methods – Euler, Milstein, and Wagner-Platen – were implemented following the book by Kloeden & Platen [1]. These methods make use of samples of the principal Wiener integrals listed above. Unfortunately, the text contains typographic errors, which were found by examining the mean, variance and covariance properties of fundamental multiple Itō and Stratonovich integrals using a testing suite that is part of the Pychastic package. The errors we found were corrected appropriately and are listed in the Appendix A.

Finally, using the jax.lax.scan functionality of the package jax, we can generate many trajectories in parallel by programmatically vectorising the code representing $a$ and $b$ provided by the user.

# 4 Examples of usage

In the following, we show a few examples of usage that take advantage of various functionalities of Pychastic.

## 4.1 First passage problems, polar random walk

In nearly all physical applications of Brownian Dynamics, the simulated properties of the systems are observed via their moments rather than individual realisations of the underlying stochastic process. In fact, in many situations with random forcing, the paths of the process are only a model of reality and cannot be directly observed, as opposed to their statistical effect. For a smooth quantity $g$ observed through its expected value $\mathbb{E}$ in thermodynamic equilibrium, it is the weak convergence rate that controls the error of $\mathbb{E}(g(X_t))$ at fixed $t$. With this in mind, it is tempting to dismiss methods with a high strong order of convergence as impractical.

A natural candidate for a counterexample are first passage times, where the answer requires a more subtle reasoning. Since the first passage problem is *path-dependent*, it would seem that strong convergence is important. However, the theorem regarding the first passage times established by Whitt [19] shows that the weak convergence of the process approximations $X_n \to X$ implies a weak convergence of the first passage times $T(X_n) \to T(X)$. In conclusion, for all popular physical applications, the weak order of convergence is the important one. In consequence, the Milstein scheme is never a good choice, since it is equivalent to the Euler scheme in terms of weak order of convergence, but is more computationally intensive and harder to implement. We illustrate the surprising result of [19] by numerical simulation in a familiar setting.

A simple analytically solvable case of the first passage problem is a two-dimensional diffusion process in the $(X, Y)$ plane with constant drift velocity $\boldsymbol{v}$ and constant diffusion coefficient $\sigma^2$ in a system with an absorbing barrier at $Y = Y_b = 2$. The problem admits an exact solution, in which the probability density function for the hitting time $t_{\text{hit}} = \text{argmin}_t(Y_t > Y_b)$ is given by

$$p_{t_{\text{hit}}}(\tau) = \frac{Y_b}{\sqrt{2\pi\sigma^2\tau^3}} \exp\left(-\frac{(Y_b - v_y\tau)^2}{2\sigma^2\tau}\right), \tag{10}$$

and so $\mathbb{E}[t_{\text{hit}}] = 1/2$.

Numerically, we calculate the sample of the first passage times and locations for $\boldsymbol{v} = (v_x, v_y) = (0, 4)$ with stopping condition $Y = Y_b = 2$ and initial condition $(X, Y) = (2, 0)$. The Itō equations for the two-dimensional diffusion driven by Wiener processes $W$ and $W'$ with drift velocities $[v_x, v_y]$ take the form of

$$dX = v_x dt + \sigma dW_t, \tag{11}$$
$$dY = v_y dt + \sigma dW'_t, \tag{12}$$

where $\sigma$ is a parameter that describes the strength of the noise compared to the drift. We first transform this problem into a different set of coordinates, for which we choose polar coordinates. While we retain the analytical solution, we acquire non-linear terms in the governing equations, which will help us study the convergence issues which arise when noise and drift terms are generally dependent on the position. Eq. (12) transformed into polar coordinates $(r, \phi)$ is expressed as

$$dr = \left(\tfrac{1}{2r} + v_y \sin\phi + v_x \cos\phi\right) dt + \sigma\cos\phi\, dW_t + \sigma\sin\phi\, dW'_t, \tag{13}$$
$$d\phi = \tfrac{1}{r}\left(v_y \cos\phi + v_x \sin\phi\right) dt - \sigma\frac{\sin\phi}{r} dW_t + \sigma\frac{\cos\phi}{r} dW'_t. \tag{14}$$

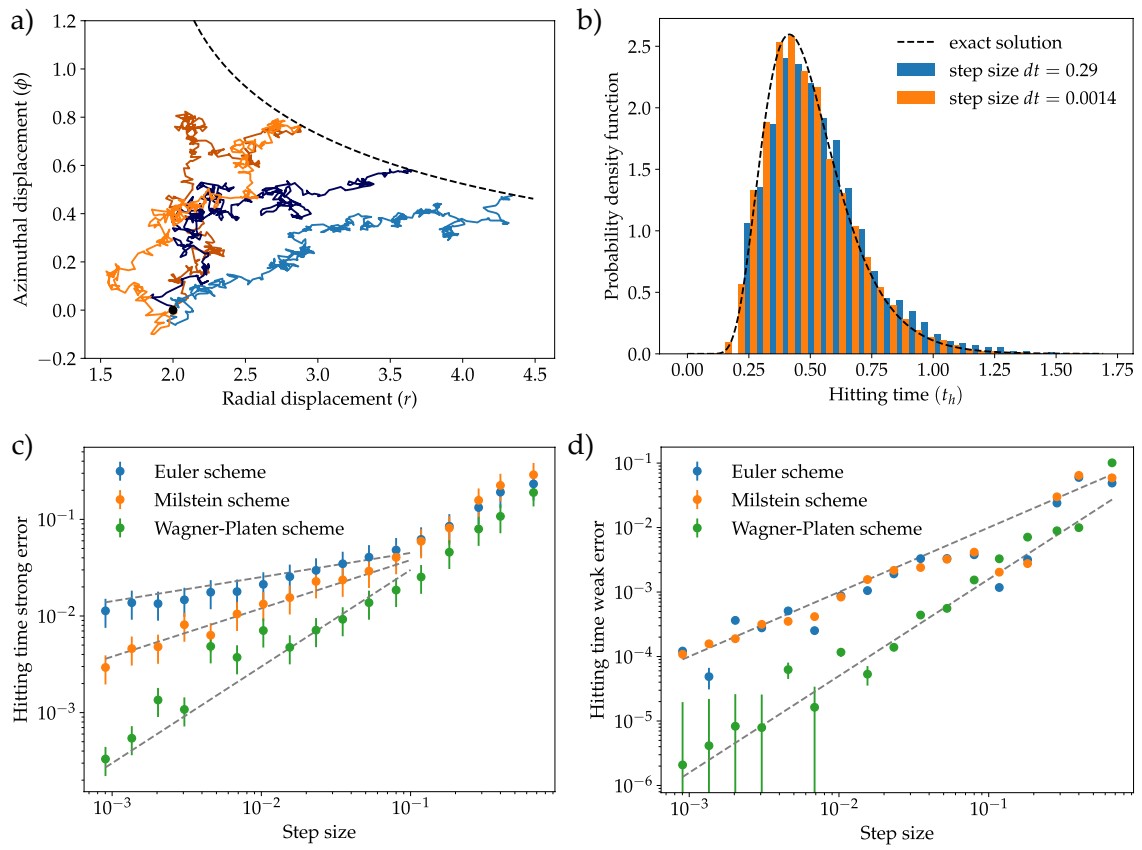

Figure 1: **a)** Four sample trajectories generated from Eq. (14) using the Euler algorithm together with the barrier $r \sin \phi = y = 2$. **b)** Distribution of hitting times for two different step sizes in the Euler algorithm together with the analytic solution of the problem. Large step sizes typically overestimate the hitting time. **c)** Strong (path-wise) error of the hitting time for different step sizes and solvers together with power-law eye guides: $dt^{1/4}, dt^{1/2}, dt$. **d)** Error of the expected value of the first passage time together with eye-guides: $dt^{1/2}, dt$. Contrary to the strong error case, here Euler and Milstein algorithms coincide by Whitt's theorem.

The transformed equations are nonlinear, coupled, with nondiagonal, noncommutative noise terms. Vector equations in the package `Pychastic` are defined by providing both the drift vector `a` and the noise matrix `b` as callable to the `SDEProblem` constructor. Denoting the configuration by $q = (r, \phi)$, we write

```python
import jaq.numpy as jnp

y_drift = 4.0
sigma = 1
y_barrier = 2

def drift(q):
    return jnp.array(
        [1 / (2 * q[0]) + y_drift * jnp.sin(q[1]), y_drift * jnp.cos(q[1]) / q[0]]
    )
```

```
11
12 def noise(q):
13     return sigma * jnp.array(
14         [[jnp.cos(q[1]), jnp.sin(q[1])], [-jnp.sin(q[1]) / q[0], jnp.cos(q[1]) / q[0
       ]]]
15     )
16
17 problem = pychastic.sde_problem.SDEProblem(
18     a=drift,
19     b=noise,
20     x0=jnp.array([2.0, 0.0]),
21     tmax=2.0,
22 )
```

The `jax.numpy` package is a functional cousin of `numpy` [20], which is the fundamental package for scientific computing with Python [21]. The functional focus of `jax.numpy` enables automatic differentiation, and thus facilitates the use of the high-order method without the need for numerical differentiation, which can be imprecise and computationally costly. We obtain the trajectories using the `solve` method of `SDESolver`.

```
1 solver = pychastic.sde_solver.SDESolver(dt=2**(-5), scheme='milstein')
2 solution = solver.solve_many(problem,n_samples,seed=0)
```

`Pychastic` supports simulating many trajectories simultaneously with the `solve_many` method. Our strategy allows for concurrency in computation of random variables and equation coefficients thanks to the `jax` package we use as back-end (which in turn relies on vectorised mathematical operations provided by the `XLA` supporting architecture of most modern processors). The `jax` package is a fusion of three capabilities: `numpy`-like API for array-based computing, functional transformations (such as vectorisation, parallelisation and automatic differentiation), and modular back-end, allowing developers to test their work with just CPU and deploy the same code later on a GPU (or TPU) capable hardware. Furthermore, we avoid notoriously slow `Python` loops by using `jax.lax.scan` routines, taking advantage of just-in-time compilation and asynchronous dispatch, avoiding the problem of global interpreter lock, which cripples imperatively coded Python programmes. A more detailed guide on the advantages (and common pifalls) of `jax` package can be found in its documentation on ReadTheDocs.

Importantly, even though the Milstein and Wagner-Platen schemes require the values of spatial derivatives of noise and drift terms, we did not have to provide them explicitly. They were calculated using an automatic differentiation procedure from the coefficient functions provided in the `SDEProblem` constructor.

We compare the results obtained with different solvers and time steps in Fig. 1. Sample trajectories starting from the point $(r, \phi) = (2, 0)$ are shown in Fig. 1a. The barrier at $y = 2$ becomes a curve in polar parameterisation. In Fig. 1 we present the resulting distribution of hitting times measured from an ensemble of $N = 1000$ trajectories with two different time-step sizes and compared to the exact solutions. Choosing a too large step size can lead to an overestimation of the typical hitting time. Having implemented three stochastic integration algorithms, we compare the strong error of estimation of the hitting time in Fig. 1c. It is a measure of the numerical error per trajectory, defined as $\langle |\tilde{\tau} - \tau| \rangle$, where $\tilde{\tau}$ is the numerical estimate of hitting time and $\tau$ is actual hitting time that can be computed using the exact solution and a particular realisation of the Wiener process. The average is taken over an ensemble of realisations of the Wiener process. For different schemes, the strong error scales differently with the step size – from the linear dependence on

$dt^{1/4}$ for the Euler scheme, through $dt^{1/2}$ for the Milstein algorithm, to $dt$ for Wagner-Platen. We note that these exponents are different (smaller) than for strong convergence at a fixed time [1]. Similarly, in Fig. 1d we present the weak error of estimation of the average value of the hitting time. The weak error is defined as $|\langle \tilde{\tau} \rangle - \langle \tau \rangle|$, and reflects the difference between the ensemble average value of hitting time estimate and the true ensemble average hitting time (computed directly from the theoretical distribution). This quantity highlights the lack of difference between Euler and Milstein schemes, in agreement with Whitt's theorem [19] stating the dependence of the expected first passage time on the weak convergence rate of the scheme only. The power laws for convergence – $dt^{1/4}$ for Euler and Milstein algorithms and $dt$ for the Wagner-Platen scheme – again have smaller exponents than for those for weak convergence at a fixed time.

## 4.2   Rotational Brownian motion, `step_post_processing` function

To highlight the ease-of-use features of the `Pychastic` package, we present the problem of rotational Brownian diffusion simulations. General BD simulations that involve rigid bodies require the application of finite rotations to diffusing objects. The resulting algorithms are widely applied to study problems in physics and biology. While resolving translational motion is straightforward, e.g. with the standard Ermak-McCammon (Euler) algorithm [22], the rotational part is more involved to simulate, because the domain of rotational motion is $SO(3)$ and this has to be taken into account when solving the equations of motion. One difficulty lies in the commonly used rotational coordinate systems, such as Euler angles [23], which contain strong singularities around the polar orientations. When curvilinear coordinates are used to describe the motion, the metric tensor gives rise to nontrivial additional terms in the equations of motion, the so-called metric or drift terms, which are frequently overlooked in rotational BD algorithms. For a summary, see Ref. [24].

To overcome these limitations, we reimplement the problem following the rotation-vector-based formulation of Evensen *et al.* [25]. We describe the angular position of the particle by the angle of rotation $\Phi$ around a unit vector $\boldsymbol{\delta}$ collinear with the axis of rotation. Generalised coordinates are encoded in the vector $\boldsymbol{q} = \Phi \boldsymbol{\delta}$. The rotational mobility matrix of the particle in the body-fixed frame is $\boldsymbol{\mu}_{\text{body}}$. We transform it into the laboratory frame by

$$\boldsymbol{\mu} = \boldsymbol{\Omega}^T \cdot \boldsymbol{\mu}_{\text{body}} \cdot \boldsymbol{\Omega}, \tag{15}$$

where $\boldsymbol{\Omega}$ is the relevant rotation matrix. We also define a velocity transformation matrix $\boldsymbol{\Xi}$ to further introduce the transformed mobility $\widehat{\boldsymbol{\mu}}$ by

$$\widehat{\boldsymbol{\mu}} = \boldsymbol{\Xi} \cdot \boldsymbol{\mu} \cdot \boldsymbol{\Xi}^T. \tag{16}$$

Now we can write the Itō SDE corresponding to the evolution of the generalised coordinates as

$$d\boldsymbol{q} = \widehat{\boldsymbol{\mu}} \cdot \left( \frac{\partial}{\partial \boldsymbol{q}} \log V \right) dt + k_B T \left( \frac{\partial}{\partial \boldsymbol{q}} \cdot \widehat{\boldsymbol{\mu}} \right) dt + \sqrt{2 k_B T} \widehat{\boldsymbol{\mu}}^{1/2} \cdot d\boldsymbol{W}. \tag{17}$$

Here, $V$ is the density of the volume element ($SO(3)$ Haar measure) with respect to the Lebesgue measure on $\mathbb{R}^3$. In the literature $\partial \log V / \partial \boldsymbol{q}$ is often called the metric force $\boldsymbol{F}^{(m)}$. Explicit expressions for $\boldsymbol{\Xi}$, $\boldsymbol{\Omega}$ and $\log V$ are given in Appendix B.

Since the evolution equation depends on the divergence of a product of two orientation-dependent matrices, writing Eq. (17) explicitly is quite cumbersome. Earlier works of [25]

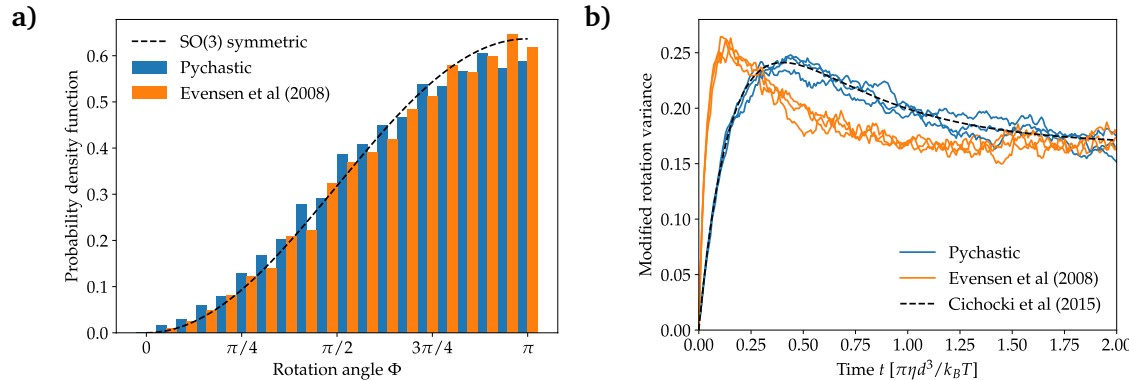

Figure 2: **a)** Distribution of rotation angles $\Phi$ after a long time in $N = 10^4$ simulations compared with the $SO(3)$ symmetric distribution $P^{eq}(\Phi)$ given by Eq. (18). **b)** Comparison of the time-dependent correlation structure arising from equation (17) for a spherical particle. Dashed lines show exact predictions from Ref. [26]. Solid, colored lines show average values of $10^3$ simulations (blue line corresponds to equation (17) and orange points correspond to Ref. [25] containing a typo in their Eq. 12). Note that even though equilibrium distributions coincide, time dependent correlation structure is different.

and [27] avoid this difficulty by approximating the gradients by sampling the transformed mobility matrix at nearby locations in the phase space. We can address this problematic term directly thanks to the automatic differentiation capabilities of the `jax` package.

```python
def metric_force(q):
    # Metric force, providing the Boltzmann distribution in equilibrium.
    phi = jnp.sqrt(jnp.sum(q ** 2))
    scale = jax.lax.cond( # Taylor expansion for the polar orientations.
        phi < 0.01,
        lambda t: -t / 6.0,
        lambda t: jnp.sin(t) / (1.0 - jnp.cos(t)) - 2.0 / t,
        phi,
    )
    return jax.lax.cond(
        phi > 0.0, lambda: (q / phi) * scale, lambda: jnp.array([0.0, 0.0, 0.0])
    )

def t_mobility(q):
    # Mobility matrix transformed to rotation vector coordinates.
    Return transformation_matrix(q) @ mobility @ (transformation_matrix(q).T)

def drift(q):
    # jax.jacobian has the differentiation index last (like mu_ij d_k) so divergence
     is contraction of the first and last axis.
    return (
    t_mobility(q) @ metric_force(q)
    + jnp.einsum("iji->j", jax.jacobian(t_mobility)(q))
    )
```

First, including the divergence of the mobility matrix in the new coordinate system is as simple

as adding a correct contraction of `jax.jacobian` of the mobility tensor. To obtain this, we use the convenient `jnp.einsum` tool. This function, similarly to its analogue `np.einsum`, takes in a tensor, represented in the memory by an array of arrays of arrays etc. with the number of levels corresponding to the tensor rank, and performs contraction described symbolically by the index notation. In our case `"iji->j"` means 'contract the first index with the last index of a rank-3 tensor to produce a vector.'

Second, the domain of the equation is $SO(3)$, which is not covered by $\mathbb{R}^3$ (since $\pi_1(SO(3)) = \mathbb{Z}_2$ [18]). As a result, some trajectories, continuous in $SO(3)$, will necessarily be discontinuous in the $\mathbb{R}^3$ parameterisation. We adopt the method proposed by Ref. [25], where after each step of integration we project on the principal value of the rotation angle. In `pychastic`, we achieve this using the `step_post_processing` capability.

```python
def canonicalize_coordinates(q):
    phi = jnp.sqrt(jnp.sum(q ** 2))
    max_phi = jnp.pi
    canonical_phi = jnp.fmod(phi + max_phi, 2.0 * max_phi) - max_phi
    return jax.lax.cond(
        phi > max_phi,
        lambda canonical_phi, phi, q: (canonical_phi / phi) * q,
        lambda canonical_phi, phi, q: q,
        canonical_phi,
        phi,
        q,
    )
solver = pychastic.sde_solver.SDESolver(dt=0.01)
trajectories = solver.solve_many(
    problem,
    step_post_processing=canonicalize_coordinates,
    n_trajectories=1000
)
```

The simulations were validated by checking the equilibrium properties and the time-dependent correlation structure. The equilibrium distribution $P^{eq}(\Phi)$ for the rotation angle $\Phi$ is given by

$$P^{eq}(\Phi) = \frac{1 - \cos\Phi}{\pi}. \tag{18}$$

To characterise the time evolution, we use the components of the modified rotation vector components $\Delta u_k(t) = -\frac{1}{2}\epsilon_{ijk}\Omega_{ij}(t)$, for which exact theoretical predictions were derived by Cichocki *et al.* [26] for an arbitrarily shaped molecule. For a spherical particle, the correlation functions are given by

$$\langle \Delta u_k(t) \Delta u_l(t) \rangle_0 = \left[\frac{1}{6} - \frac{5}{12}e^{-6D_r t} + \frac{1}{4}e^{-2D_r t}\right] \delta_{kl}^K, \tag{19}$$

where $\delta^K$ is the Kronecker delta. The diffusion coefficient for a sphere is $D_r = k_B T/\pi\eta d^3$, where $d$ is the diameter of the particle.

In Fig. 2, we present a comparison between the results obtained with an algorithm based on Eq. (17) (blue lines), and a similar method based on equations (5-12) from Ref. [25] (orange). Fig. 2a shows the equilibrium distribution of the rotation angle $\Phi$, together with the analytical prediction of Eq. (18). We note that although the equilibrium distribution predicted by both algorithms shows exact agreement with the theoretical predictions, the transformation matrix $\Xi$

in Eq. (12) of Ref. [25] contains an error. We corrected this typographical error and present the proper formulation of the equations of motion matrices in the Appendix B. The discrepancy is visible in the time-dependent correlation function in Fig. 2b. Although at long times the system tends to the same equilibrium solution for both approaches, we see the agreement of our algorithm with the theoretical predictions of Eq. (19) (dashed line) and the deviation of the orange lines. This discrepancy highlights the need for better test cases that reliably test all properties of the simulated equation, and not solely the equilibrium distribution.

As mentioned above, the algorithm based on Ref. [25] is singularity-free, contrary to approaches based on Euler angles, which contain strong singularities around the polar orientations. However, it should be noted that the implementation of the code may still face some limitations, such as the computational inability to calculate $\sin\Phi/\Phi$ for $\Phi \to 0$. For these numerically restricted cases, we implemented the Taylor expansion of $\Xi$, $\Omega$ and $\log V$, based on Ref. [27]. However, we also report some typos in this publication. Therefore, in Appendix B we provide correct formulations of the Taylor-expanded terms.

## 4.3 Bead models with hydrodynamic interactions, `pygrpy` package integration

Hydrodynamically interacting beads (with and without springs) have been used successfully in modelling the properties of elastic macromolecules [22, 28–32]. Some questions about elastic macromolecules can be answered by computing the equilibrium ensemble of conformations (with methods such as Markov Chain Monte Carlo [33]). Simulations involving hydrodynamic interactions can, on the other hand, provide access to the dynamics and answer questions, e.g., about timescales of conformational change [34], mechanisms of protein association [35], pore translocation [36–38], near-wall hindered diffusion [39], and many other dynamical processes.

The starting point is often the Hamiltonian, which describes intramolecular interactions between the constitutive subunits of the molecule, modelled by a collection of beads. The potential energy landscape can then be used to compute the interaction forces. Thanks to the `jax` autograd capabilities, the forces arising from many-body mechanical interactions can be automatically calculated from the potential energy of the system.

The computation of hydrodynamic interactions, encoded in the mobility tensors of the respective macromolecules is a more involved task. Numerous methods are available, varying in scope, degree of precision, and complexity. For a review of popular methods applied to macromolecules, see Ref. [40]. For completeness, we supplemented `pychastic` with the package `pygrpy` for the calculation of grand mobility tensors in the Rotne-Prager-Yamakawa (RPY) approximation [41, 42], generalised to beads of different sizes [43]. The procedure is a Python port of the `GRPY` package [40]. The Python package `pygrpy` simplifies the implementation of similar bead-spring simulations, both stochastic and deterministic. Furthermore, `pygrpy` is compatible with the `jax` functional paradigm and allows automatic differentiation and vectorisation.

The RPY approximation is by far the most popular method of accounting for hydrodynamic interaction in numerical models of soft matter systems [44]. The mobility tensors calculated in this way preserve positive definiteness and are divergence-free, significantly simplifying the BD algorithm [22]. In essence, RPY is a far-field approximation that includes all terms that decay slower than the inverse third power of the interparticle distances, but is less accurate at smaller distances. When the particles come close together, it is necessary to include higher-order terms of the multipole expansion [45] and lubrication corrections [46]. However, as shown by Żuk *et al.* [43], the RPY approximation can be generalised to overlapping particles, and thus it can also be used to model complex-shaped particles as conglomerates of rigidly glued overlapping spheres,

in particular to calculate hydrodynamic properties of biological macromolecules, as in the GRPY method [40].

To highlight the ease of use and interoperability of `pychastic` and `pygrpy`, we implement a benchmark problem proposed by Cichocki et al. [47]. It concerns the diffusion coefficient of an elastic "macromolecule" composed of 4 beads of radii $r_i \in \{3, 1, 1, 1\}$ joined into a string, as shown in Fig. 3a. Here, the length scale is the radius of a small bead $a$. The neighbouring beads interact directly via harmonic potentials, with the equilibrium distance $d_i = 4$ and the spring constant $k = 5.5 \, k_B T / a$, and indirectly through hydrodynamic interactions.

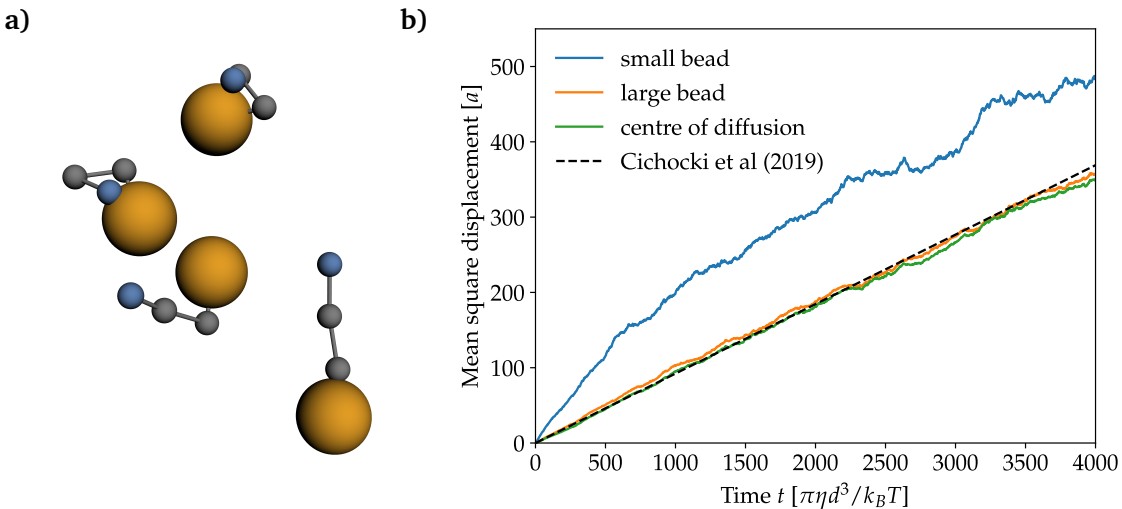

Figure 3: **a)** Representative configurations of four co-diffusing beads connected with harmonic springs at time (bottom to top) $t / \tau_d = 2, 2000, 4000, 6000$. **b)** Mean square displacement of three different tracked locations: the small bead at the end of the chain (blue), the large bead at the other end (orange) and the weighted average of all four beads with weights corresponding to the effective centre of diffusion, defined according to Ref. [47]. The dashed line corresponds to numerical results reported in [47] based on an extremely long, single-trajectory simulation.

We determine the diffusion coefficient $D$ by tracking the mean square displacement $\langle (x(t) - x(0))^2 \rangle$ (MSD) of a point $x$ of the molecule. If the observation time is long enough, $\mathrm{MSD} \approx 6Dt$. However, for shorter times, the coefficient of proportionality in the MSD($t$) curve is different and depends on the choice of the reference point on the molecule. Although biophysical experiments typically measure the long-time diffusion coefficient, numerical simulations have easier access to short-time diffusivity. To minimise the difference, one should choose a particular point, called the centre of diffusion, which can be constructed as a weighted average of the positions of beads with weights determined from the hydrodynamic mobilities of an ensemble of equilibrium configurations of the elastic molecule [47].

We use this system to test our algorithm. In Fig. 3b, we show the temporal evolution of the MSD when different reference points on the elastic molecule are chosen: the small terminal bead, the large bead, and the centre of diffusion. The time is scaled by $\tau_d = \pi \eta d^3 / k_B T$. We computed hydrodynamic interactions using the package `pygrpy`, which allows computations of mobility tensors for macromolecules composed of unequally sized and possibly overlapping spherical beads in the Rotne-Prager-Yamakawa approximation. The dashed line is the numerical result of [47] where

the diffusion coefficient was estimated using a single very long BD trajectory. Indeed, we see that tracking the centre of diffusion provides a good estimate of the long-time behaviour of the MSD. When one chooses to track one of the smaller beads or one of the beads further away from the middle of the molecule, precision decreases primarily because rotational diffusion plays a bigger role in the motion of such tracers. The strength of our algorithm is that we use many trajectories instead of a single long one, thus avoiding the need for a more complex method of calculating the diffusion coefficient, involving recursive subdivisions of the simulation interval described by Frenkel and Smit [48].

```python
radii = jnp.array([3.0,1.0,1.0,1.0]) # sizes of spheres used
n_beads = len(radii)
equilibrium_dist = 4.0
spring_constant = 5.5

def pot_ene(x): # potential energy
    locations = jnp.reshape(x,(n_beads,3))
    distance_ab = jnp.sqrt(jnp.sum((locations[0] - locations[1])**2))
    distance_bc = jnp.sqrt(jnp.sum((locations[1] - locations[2])**2))
    distance_cd = jnp.sqrt(jnp.sum((locations[2] - locations[3])**2))
    ene = 0.5*spring_constant*jnp.sum(
    ([distance_ab,distance_bc,distance_ca] - equilibrium_dist)**2
    )
    return ene

def drift(x):
    locations = jnp.reshape(x,(n_beads,3))
    mu = pygrpy.jax_grpy_tensors.muTT(locations,radii)
    force = -jax.grad(pot_ene)(x)
    return jnp.matmul(mu,force)

def noise(x):
    locations = jnp.reshape(x,(n_beads,3))
    mu = pygrpy.jax_grpy_tensors.muTT(locations,radii)
    return jnp.sqrt(2)*jnp.linalg.cholesky(mu)

problem = pychastic.sde_problem.SDEProblem(
    drift,
    noise,
    x0 = jnp.reshape(jnp.array([
    [-2.,0.,0.],
    [2.,0.,0.],
    [6.,0.,0.],
    [10.,0.,0.]
    ]),(3*n_beads,)),
    tmax = 10000.0)
```

## 5 Conclusion

We developed a novel Python package pychastic dedicated to efficient numerical solutions of SDEs. The package implements the classical truncated Taylor-Itō integrators up to strong order

$\mathcal{O}(dt^{3/2})$ providing a precise treatment of both weak (e.g. equilibrium distributions, diffusion coefficients) and strong problems. We included a set of simple test cases that provide exact reference points for testing future stochastic integration algorithms. The analysis of three-dimensional rotational Brownian motion benchmarks is particularly important because it encompasses many of the difficult aspects of SDE approaches to Brownian dynamics: divergence terms in the evolution equation, handling discontinuous trajectories (unavoidable in the case of $SO(3)$), and spurious agreements when testing only the equilibrium distribution. We hope that `pychastic` will ease future studies of Brownian dynamics problems, especially problems that involve hydrodynamic interactions. The project is open source, and we hope to encourage collaboration and its further development.

## Acknowledgements

The authors thank Piotr Szymczak for his insightful feedback.

**Funding information**    The work of ML, RW, and MB was supported by the National Science Centre of Poland (FundRef DOI: http://dx.doi.org/10.13039/501100004281) grant Sonata to ML no. 2018/31/D/ST3/02408.

## A  Appendix: Typos in integration schemes in Ref. [1]

In the book: *Numerical solution of stochastic differential equations* by Peter E. Kloeden and Eckhard Platen in equation 5.8.11 we have found the following typos:

1. In the definition of $D^p_{j1,j2,j3}$: the fourth summand should read $\zeta_{j1,r}\eta_{j3,l+r}$ instead of $j1$.

2. In the definition of $b_j$: there is a $1/\pi$ factor missing before sum.

3. In the definition of $C^p_{j1,j2}$: second term should be $+1/r\eta_{j1,r}\eta_{j2,l}$ instead of $-l/r$

## B  Appendix: Rotational Brownian motion

In equation (17) we recall the equations of rotational Brownian motion, formulated in Ref. [25]. Angular orientation is described by the vector $\boldsymbol{a} = (a_1, a_2, a_3) = \Phi\boldsymbol{\delta}$, where $\Phi$ is the angle of rotation around a unit vector $\boldsymbol{\delta}$, which corresponds to the axis of rotation. Because the expressions in Ref. [25] contain typos, here we present the proper expressions for coordinate transformation and rotation matrices $\boldsymbol{\Xi}$ and $\boldsymbol{\Omega}$:

$$\boldsymbol{\Xi} = \left(\frac{1}{\Phi^2} - \frac{\sin\Phi}{2\Phi(1-\cos\Phi)}\right)\begin{pmatrix} a_1a_1 & a_1a_2 & a_1a_3 \\ a_2a_1 & a_2a_2 & a_2a_3 \\ a_3a_1 & a_3a_2 & a_3a_3 \end{pmatrix} + \frac{1}{2}\begin{pmatrix} \frac{\Phi\sin\Phi}{1-\cos\Phi} & -a_3 & a_2 \\ a_3 & \frac{\Phi\sin\Phi}{1-\cos\Phi} & -a_1 \\ -a_2 & a_1 & \frac{\Phi\sin\Phi}{1-\cos\Phi} \end{pmatrix}, \qquad (20)$$

$$\boldsymbol{\Omega} = \frac{1}{\Phi^2}\begin{pmatrix} \Phi^2\cos\Phi & -a_3\Phi\sin\Phi & a_2\Phi\sin\Phi \\ a_3\Phi\sin\Phi & \Phi^2\cos\Phi & -a_1\Phi\sin\Phi \\ -a_2\Phi\sin\Phi & a_1\Phi\sin\Phi & \Phi^2\cos\Phi \end{pmatrix} + \frac{1-\cos\Phi}{\Phi^2}\begin{pmatrix} a_1a_1 & a_1a_2 & a_1a_3 \\ a_2a_1 & a_2a_2 & a_2a_3 \\ a_3a_1 & a_3a_2 & a_3a_3 \end{pmatrix}. \qquad (21)$$

The matrix $\Xi$ transforms the velocities from the Cartesian coordinate system to the one described by $\boldsymbol{a}$. The matrix $\boldsymbol{\Omega}$ is a simple rotation matrix. These two matrices combined allow transformation of the body-fixed mobility matrix $\boldsymbol{\mu}_{\text{body}}$ to the lab-fixed, $\boldsymbol{a}$-described mobility matrix, given as $\widehat{\boldsymbol{\mu}} = \Xi \cdot \boldsymbol{\Omega}^T \cdot \boldsymbol{\mu}_{\text{body}} \cdot \boldsymbol{\Omega} \cdot \Xi^T$.

The term $\left( \frac{\partial}{\partial \boldsymbol{q}} \log V \right)$ from Eq. (17) can be associated with the metric force $\boldsymbol{F}^{(m)}$, which guarantees the Boltzmann distribution in equilibrium and is given as

$$\left( \frac{\partial}{\partial \boldsymbol{q}} \log V \right) = \boldsymbol{F}^{(m)} = -k_B T \left( \frac{\sin(\Phi)}{1 - \cos(\Phi)} - \frac{2}{\Phi} \right) \boldsymbol{\delta}. \tag{22}$$

Due to the numerical limitations, it is sometimes inevitable to perform the Taylor expansion of the above quantities, as presented in Ref. [27]. However, we also found typos in this case. Therefore, we provide the correctly expanded matrices $\Xi$ and $\boldsymbol{\Omega}$ for $\Phi \to 0$,

$$\Xi = \frac{1}{12} \begin{pmatrix} a_1 a_1 & a_1 a_2 & a_1 a_3 \\ a_2 a_1 & a_2 a_2 & a_2 a_3 \\ a_3 a_1 & a_3 a_2 & a_3 a_3 \end{pmatrix} + \frac{1}{2} \begin{pmatrix} 2 & -a_3 & a_2 \\ a_3 & 2 & -a_1 \\ -a_2 & a_1 & 2 \end{pmatrix} + O(\Phi^2), \tag{23}$$

$$\boldsymbol{\Omega} = \begin{pmatrix} 1 & -a_3 & a_2 \\ a_3 & 1 & -a_1 \\ -a_2 & a_1 & 1 \end{pmatrix} + \frac{1}{2} \begin{pmatrix} a_1 a_1 & a_1 a_2 & a_1 a_3 \\ a_2 a_1 & a_2 a_2 & a_2 a_3 \\ a_3 a_1 & a_3 a_2 & a_3 a_3 \end{pmatrix} + O(\Phi^2). \tag{24}$$

The Taylor-expanded metric force around $\Phi \to 0$ becomes

$$\boldsymbol{F}^{(m)} = k_B T \frac{\Phi}{6} \boldsymbol{\delta} + O(\Phi^3). \tag{25}$$

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
