# Peer review of "Pychastic: Precise Brownian Dynamics using Taylor-Itō integrators in Python"

_SciPost Physics Codebases, doi:SciPost Phys. Codebases 11-r0.2 (2023) , SciPost Phys. Codebases 11 (2023)_

## Round 1 · Referee Report · Anonymous (Referee 1) · 2022-11-24

Strengths

1- The manuscript introduces an open-source python package for solving stochastic differential equations in the Ito sense. The software package includes different methods for solving the SDEs numerically and also accounts for (interaction) potentials. It therefore has the potential of becoming a new, important tool for the statistical physics and soft matter community. 2- A particular feature of the python package is the implementation of the hydrodynamic mobility of interacting spheres. This could be very useful for the study of polymer problems. 3- Overall, the paper is well written and nicely introduces the package with three hands-on examples.

Weaknesses

1- The authors introduce the hydrodynamic mobility of interacting beads in terms of the Rotne-Prager-Yamakawa approximation. It would be helpful if the authors could provide some more details on the approximation, its range of validity, and potential issues. In particular, for spheres that are close together, the approximation is expected to break down. The authors could comment on this and point towards alternative methods. 2- The authors discuss strong and weak errors (in Fig.1, for example). It would be helpful if they could provide definitions for the aforementioned quantities. 3- In the conclusions, the authors could provide some perspectives about future developments and comment on many-particle-systems. For example, LAMMPS has been proven to be a useful and efficient computational method also for BD simulations. Have the authors used their software to study many-particle-systems (other than the one in the paper)? In what sense do the authors expect that their algorithm is more useful (e.g., in terms of efficiency, practicality,...) in comparison with existing approaches? Are their specific systems, where the new algorithm could be particularly useful?

Report

Let me note that, in my opinion, the manuscript does not bring forward any new physical concept/ groundbreaking discovery /new research direction... and hence does not meet any of the criteria of SciPost Physics. However, it does introduce a new computational tool for the statistical physics and soft matter communities, which can be very useful for future research and does deserve therefore publication. I would recommend to publish the paper in a more specified journal, such as: SciPost Physics Codebase.

Requested changes

Here are some minor comments and typos: - The authors may want to add references to the first paragraph in the introduction. -Perhaps I have missed it, but the authors may want to refer to their open source code somewhere in the paper. -2nd page, 2nd paragraph: '...which for a fluid of fluid of...' -> '...which for a fluid of...' - 2nd page, 2nd paragraph: introduce diffusivity D - 2nd page, 3rd paragraph: '...led to the description diffusive processes...' -> '...led to the description of diffusive processes...' - 3rd page, 2nd paragraph: 'These constitute relatively recent development...'-> 'These constitute relatively recent developments...' - 3rd page, 3rd paragraph: '...in which the packages greatly facilitates...'-> '...in which the packages greatly facilitate...' - 3rd page, 4th paragraph: '...[10,11]. Langevin equation can...' -> '...[10,11]. The Langevin equation can...' -4th page, after Eq.(3): '(provided $D$ is constant and metric tensor...)'-> '(provided $D$ is constant and the metric tensor...)' -page 13, last paragraph: 'we show the temporal evolution of MSD...' -> 'we show the temporal evolution of the MSD'

  • validity: good
  • significance: good
  • originality: good
  • clarity: high
  • formatting: excellent
  • grammar: good

Author:  Radost Waszkiewicz  on 2022-12-21  [id 3169]

(in reply to Report 1 on 2022-11-24)
Category:
answer to question
correction

Response to Reviewer 1

Strengths

  1. The manuscript introduces an open-source python package for solving stochastic differential equations in the Ito sense. The software package includes different methods for solving the SDEs numerically and also accounts for (interaction) potentials. It therefore has the potential of becoming a new, important tool for the statistical physics and soft matter community.

  2. A particular feature of the python package is the implementation of the hydrodynamic mobility of interacting spheres. This could be very useful for the study of polymer problems.

  3. Overall, the paper is well written and nicely introduces the package with three hands-on examples.

We thank the Referee for these kind words. We hope that our software package will serve the community as an easy-access tool for Brownian Dynamics simulations.

Weaknesses

  1. The authors introduce the hydrodynamic mobility of interacting beads in terms of the Rotne-Prager-Yamakawa approximation. It would be helpful if the authors could provide some more details on the approximation, its range of validity, and potential issues. In particular, for spheres that are close together, the approximation is expected to break down. The authors could comment on this and point towards alternative methods.

The Rotne-Prager-Yamakawa approximation was simply chosen as an example and we agree with the reviewer that its accuracy decreases with separation between beads. The RPY approximation is the simplest scheme for the evaluation of hydrodynamic interactions between spherical particles that preserves positive definiteness of the mobility matrix, required by the fluctuation-dissipation theorem. It is straightforward to implement and non-singular and continuously differentiable for any separation of beads, even when they overlap. This is particularly useful when modelling rigid conglomerates that are represented as a collection of beads glued together. We have revised that part of the manuscript to address this remark.

  1. The authors discuss strong and weak errors (in Fig.1, for example). It would be helpful if they could provide definitions for the aforementioned quantities.

Following this suggestion we have added appropriate definitions in relevant parts of the manuscript.

3.In the conclusions, the authors could provide some perspectives about future developments and comment on many-particle-systems. For example, LAMMPS has been proven to be a useful and efficient computational method also for BD simulations. Have the authors used their software to study many-particle-systems (other than the one in the paper)? In what sense do the authors expect that their algorithm is more useful (e.g., in terms of efficiency, practicality,...) in comparison with existing approaches? Are their specific systems, where the new algorithm could be particularly useful?

As outlined in the manuscript, correct modelling of rotational diffusion is a challenge, even for a single particle of arbitrary shape. From the programming side the biggest obstacle is dealing with the gradient of the mobility matrix terms. These issues can sometimes be avoided - for example in case of sphere or ellipsoid far from other bodies we can evaluate the change of orientation in the body frame and the equations simplify substantially. This is the approach of Brownian dynamics in LAMMPS package, as outlined in docs.lammps.org/fix_brownian. Such an approach is often sufficient, as it correctly reproduces equilibrium distribution and can be used as some approximation of time-dependent properties (as shown in the rotational diffusion section of the manuscript). However, if a more detailed resolution of the dynamics is needed, one needs tools better suited for this purpose . This need partly motivated the creation of Pychastic. With this package, it is stragihtforward to implement the spatial variation of particle diffusivity, needed e.g. to correctly resolve near-wall Brownian motion. This was not possible in LAMMPS packages.

Report

Let me note that, in my opinion, the manuscript does not bring forward any new physical concept/ groundbreaking discovery /new research direction... and hence does not meet any of the criteria of SciPost Physics. However, it does introduce a new computational tool for the statistical physics and soft matter communities, which can be very useful for future research and does deserve therefore publication. I would recommend to publish the paper in a more specified journal, such as: SciPost Physics Codebase.

We thank both reviewers and editor for bringing SciPost Physics: Codebases to our attention. It is indeed better suited for our manuscript.

Requested changes

Here are some minor comments and typos:

The authors may want to add references to the first paragraph in the introduction.

Thank you. We have added references to relevant review and outlook papers.

2nd page, 2nd paragraph: introduce diffusivity D

We have clarified this part of the manuscript. Diffusivity is now defined below eq. (2).

Perhaps I have missed it, but the authors may want to refer to their open source code somewhere in the paper.

We have provided relevant links to the package and to its documentation.

Typographic & grammatical errors:

2nd page, 2nd paragraph: '...which for a fluid of fluid of...' → '...which for a fluid of...' 2nd page, 3rd paragraph: '...led to the description diffusive processes...' → '...led to the description of diffusive processes...' 3rd page, 2nd paragraph: 'These constitute relatively recent development...'→ 'These constitute relatively recent developments...' 3rd page, 3rd paragraph: '...in which the packages greatly facilitates...'→ '...in which the packages greatly facilitate...' 3rd page, 4th paragraph: '...[10,11]. Langevin equation can...' → '...[10,11]. The Langevin equation can...' 4th page, after Eq.(3): '(provided D is constant and metric tensor...)'→ '(provided D is constant and the metric tensor...)' page 13, last paragraph: 'we show the temporal evolution of MSD...' → 'we show the temporal evolution of the MSD'

We thank the reviewer for listing these typos. We have fixed all of them in the manner suggested by the reviewer.

---

## Round 1 · Referee Report · Anonymous (Referee 2) · 2022-11-25

Report

This is my reviewer's report on the manuscript referenced above, which is being considered for publication in the SciPost family of journals. My recommendation is to swiftly publish this manuscript in SciPost CODEBASES, once the Authors have considered the optional changes suggested below.

The considered manuscript describes the implementation in Python of algorithms for the numerical simulation of Brownian motion using the Ito formalism. The Authors first motivate the relevance of numerical analyses of Brownian motion (Sec. 1), and compare the Langevin, Fokker-Planck, and Ito viewpoints (Sec. 2). Then, they describe their numerical package Pychastic and its implementation, including key parts of their code in the text (Sec. 3). Finally (Sec. 4), they demonstrate usage on three physical applications of increasing complexity: a 2D first passage problem (§4.1), an example involving rotation (§4.2), and hydrodynamically interacting beads (§4.3).

The authors make no claim to novelty in the implemented algorithms or in the considered applications, as clearly stated in the text (§3.1 and §4.1-3). The strongsuit of this manuscript is that it provides an excellent example of the recent functional programming features in Python. The whole paper is very accessible. The Authors' examples are extensively worked out and provide ample physical motivation. Therefore, I recommend swift publication in Scipost CODEBASES.

The Authors may consider the following optional suggestions:

  1. I suggest including a brief discussion of functional programming and its advantages, a description of the JAX library which the Authors' code relies on, and a reference towards its documentation.

  2. A comparison of the speeds of the three packages would be welcome. For instance, may the Julia package DifferentialEquations.jl easily be called from Python? If so, how does its efficiency compare with the Authors' native Python solution?

  • validity: -
  • significance: -
  • originality: -
  • clarity: -
  • formatting: -
  • grammar: -

Author:  Radost Waszkiewicz  on 2022-12-21  [id 3168]

(in reply to Report 3 on 2022-11-25)
Category:
answer to question
correction

Response to Reviewer 3

This is my reviewer's report on the manuscript referenced above, which is being considered for publication in the SciPost family of journals. My recommendation is to swiftly publish this manuscript in SciPost CODEBASES, once the Authors have considered the optional changes suggested below.

We thank the reviewer for the kind words and for bringing SciPost Physics: Codebases to our attention. It is indeed better suited for our manuscript.

The considered manuscript describes the implementation in Python of algorithms for the numerical simulation of Brownian motion using the Ito formalism. The Authors first motivate the relevance of numerical analyses of Brownian motion (Sec. 1), and compare the Langevin, Fokker-Planck, and Ito viewpoints (Sec. 2). Then, they describe their numerical package Pychastic and its implementation, including key parts of their code in the text (Sec. 3). Finally (Sec. 4), they demonstrate usage on three physical applications of increasing complexity: a 2D first passage problem (§4.1), an example involving rotation (§4.2), and hydrodynamically interacting beads (§4.3).

The authors make no claim to novelty in the implemented algorithms or in the considered applications, as clearly stated in the text (§3.1 and §4.1-3). The strongsuit of this manuscript is that it provides an excellent example of the recent functional programming features in Python. The whole paper is very accessible. The Authors' examples are extensively worked out and provide ample physical motivation. Therefore, I recommend swift publication in Scipost CODEBASES.

The Authors may consider the following optional suggestions: 1. I suggest including a brief discussion of functional programming and its advantages, a description of the JAX library which the Authors' code relies on, and a reference towards its documentation.

We have added a description of the package and a reference to its documentation, along with a brief comment on the advantages of functional programming.

2. A comparison of the speeds of the three packages would be welcome. For instance, may the Julia package DifferentialEquations.jl easily be called from Python? If so, how does its efficiency compare with the Authors' native Python solution?

We regret to say that the comparison of speeds of the three packages is beyond the scope of this paper. Given the lack of expertise of the authors in Julia, an incomplete comparison between Mathematica and Python could be performed. In our view, one of the main strengths of Pychastic is the use of a programming ecosystem that is increasingly popular, and therefore could become a practical tool for stochastic simulations for many researchers. We feel that this wide availability provides added value even if the simulation speeds are comparable to existing packages which, however, require the use less popular programming languages.

---

## Round 2 · Referee Report · Anonymous (Referee 1) · 2022-12-22

Report

The authors have satisfactorily addressed my comments and I can therefore recommend publication of the manuscript in SciPost Physics Codebases.

---

## Round 2 · Referee Report · Anonymous (Referee 2) · 2023-1-5

Report

This is my second reviewer's report on the manuscript referenced above. I thank the Authors for their reply. I stand by my previous recommendation, namely, a swift publication in SciPost CODEBASES.

---

## Round 2 · Author Response

Following the recommendations from two Reviewers, we have revised the manuscript and submit it to the more appropriate SciPost Physics Codebases journal. We hope that with these changes the paper is now suitable for publication.

---

## Round 2 · List of Changes

The list of (minor) changes has been published in replies to Reviewers 1 and 3 posted with the original submission.

---

## Editorial Decision

published